# The intronic branch point sequence is under strong evolutionary constraint in the bovine and human genome

Naveen Kumar Kadri [1✉], Xena Marie Mapel[1] & Hubert Pausch [1]

The branch point sequence is a cis-acting intronic motif required for mRNA splicing. Despite their functional importance, branch point sequences are not routinely annotated. Here we predict branch point sequences in 179,476 bovine introns and investigate their variability using a catalogue of 29.4 million variants detected in 266 cattle genomes. We localize the bovine branch point within a degenerate heptamer "nnyTrAy". An adenine residue at position 6, that acts as branch point, and a thymine residue at position 4 of the heptamer are more strongly depleted for mutations than coding sequences suggesting extreme purifying selection. We provide evidence that mutations affecting these evolutionarily constrained residues lead to alternative splicing. We confirm evolutionary constraints on branch point sequences using a catalogue of 115 million SNPs established from 3,942 human genomes of the gnomAD database.

[1] Animal Genomics, ETH Zürich, Universitätstrasse 2, 8092 Zürich, Switzerland. ✉email: naveen.kadri@usys.ethz.ch

Noncoding sequences of the genome are an important source of genetic variation. Transcriptome analyses in large cohorts of humans, model organisms, and livestock have revealed regulatory elements in noncoding sequences that modulate gene expression and alternative splicing[1,2]. Such regulatory elements, in particular expression and splicing QTL, contribute disproportionately to the heritability of complex traits and diseases[3].

Pre-mRNA splicing by the spliceosome is evolutionarily conserved across eukaryotes[4]. The spliceosome complex consists of five small ribonucleic protein particles (snRNP), namely U1, U2, U4, U5, and U6 and a multitude of other proteins. These subunits of the spliceosome are sequentially recruited and assembled anew[5] at the intron-exon boundary to remove noncoding sequences (introns) and ligate coding sequences (exons) together. Motifs necessary for recruitment and assembly of the spliceosome are contained in both intronic and exonic sequences[6]. Cis-acting intronic elements include the highly conserved splice acceptor and donor sites at the exon-intron boundaries, the branch point sequence and a polypyrimidine tract between the branch point and the 3′ splice acceptor site[7].

The branch point is contained within a degenerate intronic heptamer called the branch point sequence[8]. In the vast majority of introns, this motif resides between 18 and 37 bases upstream of the 3′ splice site[9–11]. During pre-mRNA splicing, the branch point sequence undergoes base-pairing with a conserved 6-bp motif (GUAGUA) of the U2 snRNA. All residues of the heptamer except the branch point undergo base-pairing[12]. The unpaired branch point, usually adenine, bulges out of the U2 snRNA-pre-mRNA duplex. An attack of the unpaired branch point nucleotide to the 5′ splice site, frees the 5′ exon, and forms a lariat intermediate. Subsequently, the free 5′ exon attacks the 3′ splice site resulting in exon ligation, and release and degradation of the intron lariat[5].

Mutations affecting the branch point sequence may compromise the assembly of the spliceosome, leading to dramatic changes in splicing and gene expression with disease consequences[10,13,14]. Despite their functional importance, branch point sequences are not systematically studied because they are not annotated in gene transfer files. Thus, widely used tools to predict functional consequences of genetic variants such as Ensembl's Variant Effect Predictor[15] are largely blind to branch point sequences.

Intronic branch point sequences have been identified experimentally in humans and model organisms[16,17]. Computational methods leveraging the features of a high confidence set of few experimentally detected branch point sequences enable systematic assessment of this intronic motif at a genome-wide scale[18–20]. Considering pre-mRNA splicing is highly conserved across species[4], we hypothesized that the computational prediction of branch point sequences is also possible in cattle, for which experimentally validated branch point sequences are not yet available. Thus, we herein predict and characterize intronic branch point sequences in the bovine genome. Utilizing a variant catalogue established from 266 whole-genome sequenced cattle, we show that evolutionary constraints are stronger on the intronic branch point than coding sequences. We uncover similar constraints on the human branch point sequence, suggesting that this intronic motif is under extreme purifying selection in the mammalian genome. Further, using transcriptome data from 76 bulls, we show that variants in branch point sequences are associated with alternative splicing events.

## Results

### The splice sites are more constrained than the coding sequence.
To quantify the evolutionary constraint on functionally important sequence elements in the bovine genome, we extracted the coordinates of nine coding and noncoding genomic features from the annotation (Ensembl version 104) of the bovine reference genome[21]. Our analysis included 35,848 transcripts from 20,785 genes with 218,245 exons and 190,203 introns, as well as 17,759 3′ UTR, 25,490 5′ UTR, 24,217 stop codons, 23,583 start codons, 190,203 3′ splice sites, and 190,203 5′ splice sites. Using a variant catalogue established from whole-genome sequencing of 266 cattle from 11 breeds (Fig. 1a, Supplementary Data 4) sequenced to an average 16.28 (±6.56)-fold coverage, we assessed the number of polymorphic sites overlapping these features.

From the variant catalogue, we considered 29,227,950 biallelic and 190,200 multiallelic SNPs. On average, we found a variable site every 85 base pairs (assuming an autosomal genome length of 2,489,368,272), i.e., 1.18 variable sites per 100 base pairs. Genic features (introns, UTRs, splice sites, exons, start codons, and stop codons) were less and intergenic regions were more variable than the genome-wide average. The splice sites were the least variable feature with 0.13 and 0.18 variants per 100 bp, respectively, at 3′and 5′ ends (Fig. 1b, Supplementary Data 5). In exons, we observed 0.81 variants per 100 bp. Features with low variation were generally enriched for rare variants (Fig. 1c, Supplementary Data 6).

Next, we, assessed constraints on nucleotides surrounding (±21 bp) the two splice sites. The number of polymorphic sites detected varied considerably at the exon-intron boundaries (Fig. 2a, Supplementary Data 7). The two splice sites with conserved GT and AG sequence motifs at 5′ and 3′ ends of introns, respectively, harbored 77 and 84% less variation than exonic nucleotides. Intronic nucleotides nearby the splice sites (i.e., the first eight and last twelve intronic nucleotides at the 5′ and 3′ splice site, respectively) were less variable than intronic positions on average. The first three intronic nucleotides next to the 5′ splice site were strongly depleted for mutations. A consensus sequence motif "GTAAG" derived from nucleotides overlapping 190,203 exon-intron boundaries indicated that these positions are conserved (Fig. 2b). At the splice acceptor site, the intronic nucleotide 2 bp upstream of the 3′ splice site was more variable than its surrounding nucleotides. The consensus sequence motif derived from 190,203 intron-exon boundaries also indicates that this position is not conserved (Fig. 2c). The relatively low variability at intronic positions upstream the 3′ splice site is likely due to constraints on the polypyrimidine tract.

In the exonic bases, the first and second positions of nucleotide triplets (group of three consecutive nucleotides) were more strongly depleted for variation than the third position reflecting wobble at the third position of the triplet. This periodic pattern is visible even though only a subset (52%) of exons starts with the first base of the coding triplet and ends with the third base of the coding triplet (Fig. 2a). We found 0.79, 0.72, and 0.95 variable sites per 100 bp at nucleotides overlapping the first, second, and third position, respectively, of coding triplets (Supplementary Fig. 1). The last exonic position preceding the 5′ splice site, however, was an exception to this periodic pattern, probably due to the contribution of a conserved guanine in canonical splicing (Fig. 2b).

### Prediction of bovine branch point sequences.
The coordinates of splice sites are accessible from gene annotation files. However, other intronic features involved in the recognition and assembly of the spliceosome, such as the branch point sequence are not annotated. To make this feature amenable to our analysis, we extracted the sequences of 179,476 introns and predicted putative branch points with the BPP software package[18] (Supplementary Data 1).

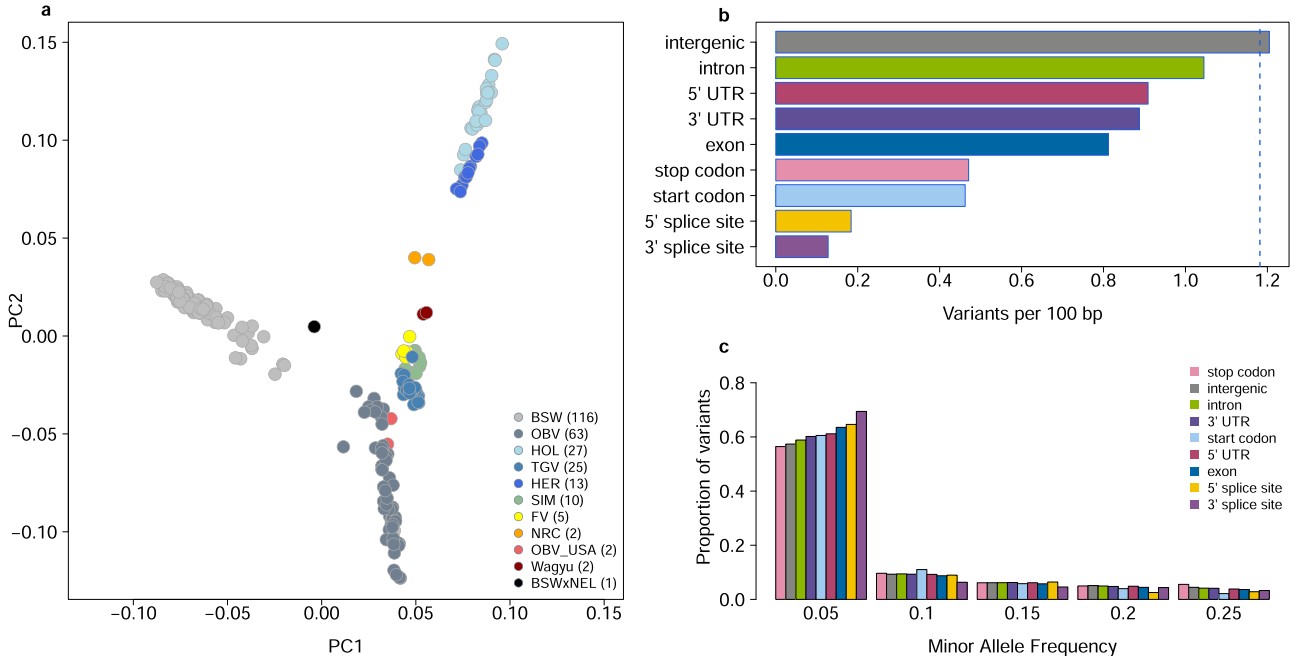

**Fig. 1 Constraints on well-annotated features uncovered from a large bovine variant catalogue.** Plot of the first two principal components (PC) of genomic relationship matrix among 266 cattle used to establish the variant catalogue. Individuals are colored according to breeds (BSW Brown Swiss, OBV Original Braunvieh, HOL Holstein, TGV Tyrolean Grey, HER Hereford, SIM Simmental, FV Fleckvieh, NRC Norwegian Red Cattle, NEL Nellore). Values in parentheses indicate the number of samples per breed (**a**). Variation (**b**) and allele frequency spectrum (**c**) within nine annotated genomic features. Variation is expressed as the number of variable nucleotides per 100 bp of the feature. The blue dotted line indicates the average genome-wide variation. Difference in variation was statistically significant (Fisher's exact test; $p < 3.87 \times 10^{-5}$) between all pairs of features except between start and stop codon.

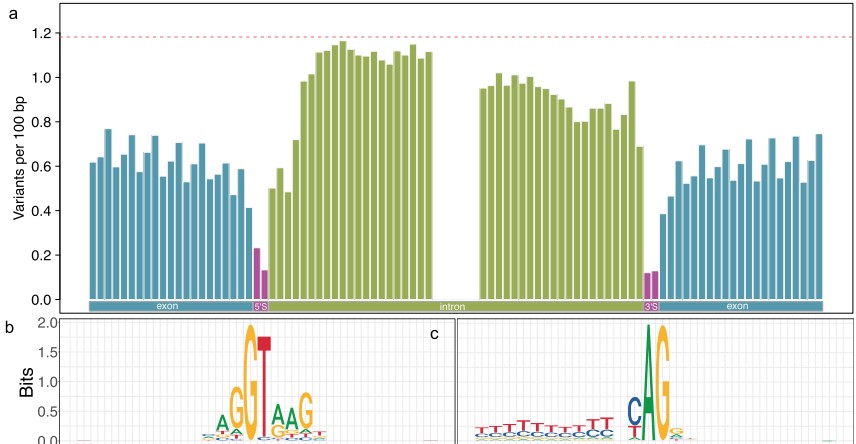

**Fig. 2 Evolutionary constraint on splice sites.** Variation observed at 21 bases up- and downstream of the 3′ and 5′ splice sites (**a**). The red dotted line represents the average genome-wide variation. The conservation of nucleotides at 5′ (**b**) and 3′ (**c**) splice sites is shown in sequence logo plots.

Intron length ranged from 22 to 1,196,801 with a median of 1322 bp. A vast majority ($n = 172,576$; 96.15%) of the introns had canonical GT and AG splice sites at 5′ and 3′ ends, respectively. The second most frequent ($n = 5608$; 3.12%) type of introns had GC and AG splice sites at 5′ and 3′ ends, respectively. The U12-type introns with AT and AC splice sites that are spliced by minor spliceosomes were exceedingly rare ($n = 127$, 0.07%). A few introns ($n = 1165$, 0.65%) had other splice sites.

The branch points resided between 14 and 145 bp upstream of the 3′ splice site with a mean distance of $28.8 \pm 11.2$ bp (Fig. 3, Supplementary Data 1). The predicted branch point was predominantly adenine (98.64%, $n = 177,035$). Cytosine (1.28%, $n = 2297$,) or thymine (0.08%, $n = 144$,) residues were rarely predicted as branch points. A thymine residue was highly

conserved (91.56%, $n = 164,329$) at the 4th position of the heptamer. The heptamer preferentially contained purines (69.89%) at the 5th, and pyrimidines at the 3rd (70.33%) and 7th (70.05%) position.

The vast majority (90.47%, $n = 162,377$) of the heptamers contained a canonical "TNA" motif with thymine and adenine residues at positions 4 and 6. Heptamers with cytosine and adenine residues at position four and six ("CNA"), respectively, were the second most frequent (7.85%, $n = 14,083$) class among the predicted branch point sequences.

Heptamers with a "TNA" motif were more distant to the 3′ splice site than those with other motifs (29.16 bp vs. 25.74 bp between the branch point residue and the 3′ splice site; $t$ test: $p < 1.0 \times 10^{-323}$). Further, branch points were more distant from

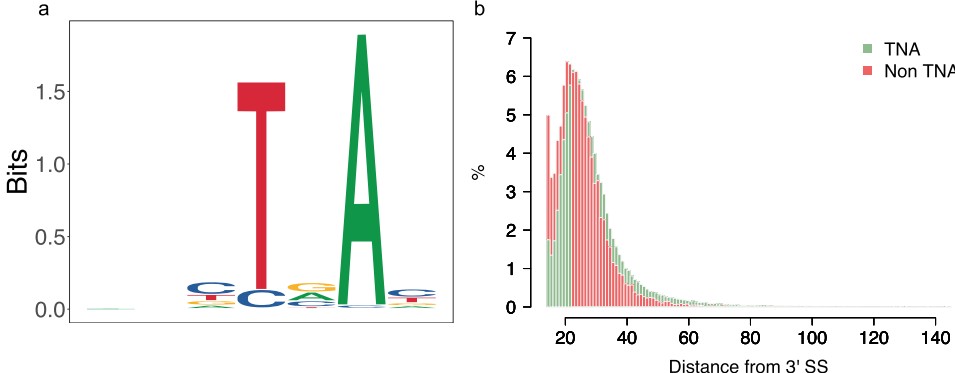

**Fig. 3 Characteristics of the branch point sequence in bovine introns.** Sequence logo plot generated from 179,476 branch point sequences identified in the bovine genome (**a**). Distribution of the distance between the predicted branch point and the 3′ splice site for canonical (TNA) (green) and noncanonical (non-TNA) (red) branch point sequences (**b**).

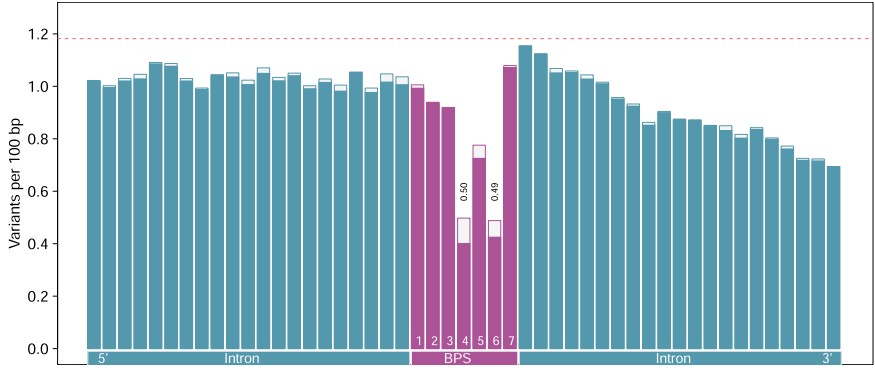

**Fig. 4 Local variation at the bovine branch point sequence (BPS).** The height of the bars indicates the variation (variants per 100 bp) surrounding 179,476 branch points while the solid bars indicate the variation surrounding a subset of 162,377 branch points encompassed by heptamers with a canonical "TNA" motif. The dotted horizontal red line indicates the average genome-wide variation.

the 3′ splice site in canonical (GT-AG type) than other introns (28.92 bp vs. 26.57 bp, *t* test: $p = 1.53 \times 10^{-70}$). Heptamers with a "TNA" motif were more frequent in GT-AG type than other introns (Supplementary Fig. 2; 91.20 vs. 72.39%; Fisher's exact test: $p < 1.0 \times 10^{-323}$). Conversely, heptamers with the noncanonical "CNA" motif were more frequent (19.87 vs. 7.37%; Fisher's exact test: $p = 4.14 \times 10^{-232}$) in noncanonical introns.

**Evolutionary constraint on the bovine branch point sequences.** Next, we investigated the variation within the predicted branch point sequences and surrounding (±21) nucleotides. Overall, nucleotides within the heptamer were less variable than the ($n = 42$) surrounding intronic nucleotides (Fig. 4, Supplementary Data 1). However, the constraint on the individual bases varied considerably within the heptamer. The reduced overall variability of the heptamer was primarily due to a striking depletion of variants at the predicted branch point (0.49 variants per 100 bp) and the residue at position 4 (0.50 variants per 100 bp) of the heptamer. Notably, these two bases were less variable than the exonic sequence (cf. Fig. 1b). The 5th position of the heptamer was also less variable (0.78 variants per 100 bp) than the intronic nucleotides surrounding the heptamer. However, the positions 1, 2, 3, and 7 of the heptamer (0.92–1.08 variants per 100 bp) were as variable as adjacent intronic nucleotides towards the 5′ end of the introns. The reduced variability of intronic sequence towards the 3′ end is likely due to the presence of an evolutionarily constraint polypyrimidine tract.

The depletion of variation at positions four and six was more pronounced within heptamers that contained the canonical "TNA" motif. Within these heptamers, the thymine residue at position four (0.40 variants per 100 bp) was less variable than the branch point adenine (0.42 variants per 100 bp). The difference, however, was not statistically significant (Fisher's exact test: $p = 0.30$).

We then compared branch point sequence prediction from BPP to predictions from two other tools LaBranchoR[19] and branchpointer[20] (Supplementary Note 2). Unlike BPP, the two other tools pose restriction on length of the intronic sequence hence the predictions were available only for a subset of 177,668 and 178,559 introns from LaBranchoR and branchpointer, respectively. The pattern in local variation around the branch point sequences predicted with these tools mirrored the pattern observed for predictions from BPP (Supplementary Figs. 3 and 4). The overlap in predictions between BPP and the two other tools was ~60%. The predicted branch point across the three tools matched in 93,855 (52.8%) introns. We observed the highest overlap in predictions between LaBranchoR and branchpointer (75%; $n = 133,972$). For the three tools considered, the prediction scores for the 93,855 matching branch points were significantly higher and were weakly, yet significantly, negatively correlated with the variability of the heptamer (Supplementary Note 2).

The depletion of variation at positions 4 (0.34 variants per 100 bp) and 6 (0.31 variants per 100 bp) was even stronger for a subset of 93,855 branch point sequences that were placed at the same intronic position by the three tools (Supplementary Fig. 5). The difference in variability between the two conserved positions of the heptamer was, however, not statistically significant (Fisher's exact test: $p = 0.2$). Among 93,885 introns for which

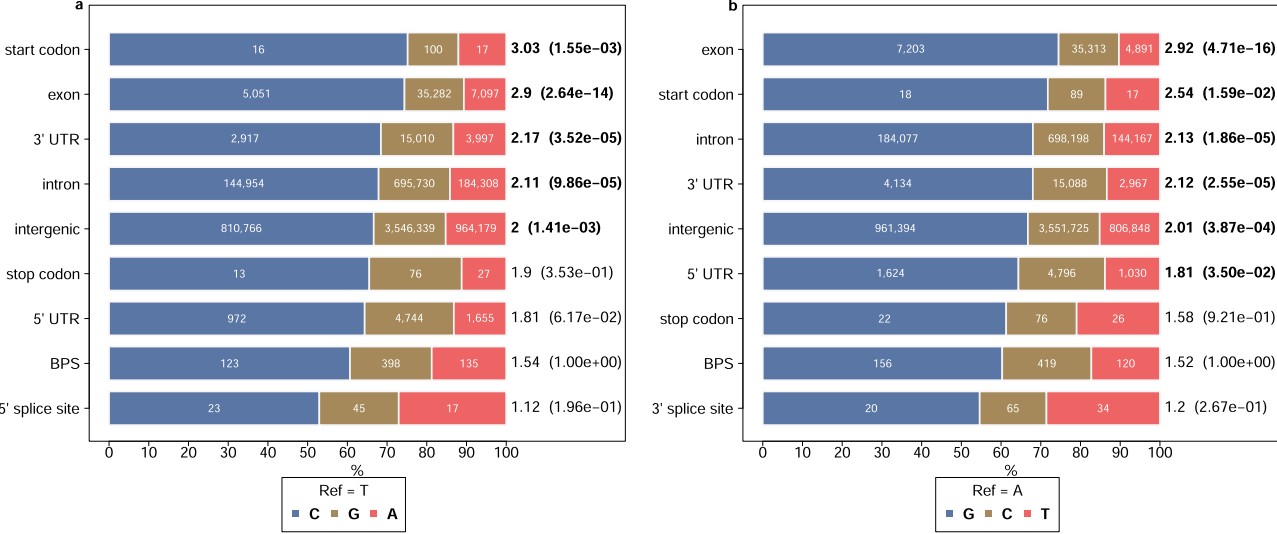

**Fig. 5 Ti/Tv ratio of different genomic features.** SNP mutation types of thymine at position 4 (**a**) and adenine at position 6 (**b**) within canonical ("TNA") branch point sequence motifs and other genomic features. The widths of differently colored bars indicate the proportion of substitution to the other three nucleotides. The number of substitutions observed in our variant catalogue is given in the middle of each differently colored bar. The Ti/Tv ratio for the features is indicated at the end of the bars. Values in the parenthesis are *p* values (<0.05 indicated in bold text) from pairwise Fisher's exact tests comparing the Ti/Tv ratio of nucleotides within the BPS and other genomic features. For splice sites, analysis was limited to a splice site where the canonical sequence motif (GT and AG at 5′ and 3′ splice sites respectively) contains the studied residues.

all tools identified the same branch point sequence, the predicted heptamer contained a canonical "TNA" motif for 90,776 introns. The depletion of variants in these heptamers was similar at the branch point and the fourth position of the heptamer (0.29 variants per 100 bp).

The widely accepted model for splice site recognition suggests that the first AG dinucleotide downstream of the branch point sequence is selected as splicing site[10,22]. The absence of AG dinucleotides between the branch point sequence and splicing acceptor site, the so-called AG exclusion zone (AGEZ), is thus necessary for accurate splice site selection[23]. We, therefore, tabulated point mutations that introduce AG dinucleotides in 178,690 bovine introns with canonical AG splice acceptor site and tested if such mutations are depleted in the AGEZ (Supplementary Note 3). Point mutations that insert AG dinucleotides (mutations to adenine at nucleotides one bp upstream guanine and mutations to guanine at nucleotides one bp downstream of adenine residues) were indeed strongly depleted in the AGEZ. Such mutations were only half as frequent in the AGEZ than other intronic regions (Fisher's exact test: $p < 1.8 \times 10^{-155}$).

**The conserved thymine and adenine residues are enriched for transversions.** Next, we investigated the mutational pattern at the canonical "TNA" motif. Specifically, we counted single-nucleotide substitutions of thymine at position 4 (to adenine, cytosine, and guanine), and adenine at position 6 (to cytosine, guanine, and thymine) of the heptamer, and compared it to substitutions of adenine and thymine residues in other genomic features.

Within all genomic features investigated, transitions were more frequent than transversions at both adenine and thymine residues. However, the transition to transversion (Ti/Tv) ratio varied in the features studied (Fig. 5, Supplementary Data 8). Our variant catalogue contained 656 and 695 SNPs, respectively, affecting thymine and adenine residues at positions 4 and 6 of canonical heptamers, with a Ti/Tv ratio of 1.54 and 1.52, respectively. The Ti/Tv ratio was lower at these thymine and adenine residues than within the other features tested except the two splice sites. Importantly, their Ti/Tv ratio was significantly

(Fisher's exact test: $p < 9.9 \times 10^{-5}$) lower compared to other intronic thymine (2.11) and adenine (2.13) nucleotides.

**A conserved thymine residue is the least variable site in human branch point sequences.** Following the approach introduced earlier, we considered 115,640,370 high-quality SNPs segregating in 3942 samples of diverse ancestry to estimate the variation in various functional features of the human genome. A larger number of sequenced human samples provided a more comprehensive catalogue of SNPs including many that segregated at very low minor allele frequency. Thus, the average genome-wide variation was more than threefold higher (4.21 variable sites per 100 bp) in the human than cattle data. The variation within the genomic features relative to the genomic average, however, was largely consistent with the estimates obtained in the cattle data (Supplementary Note 1).

Next, we predicted branch point sequences in 201,832 human introns using the BPP software[18]. The predicted branch point was predominantly adenine (98.87%), and it was placed 29.3 ± 11.9 bp upstream of the 3′ splice site. The pattern of variability for nucleotides overlapping the heptamer resembled our earlier findings in the bovine dataset. The branch point and the position 4 of the heptamer harbored 35.2 and 40.6% lower variation than the coding sequence, respectively. The position 4 of the heptamer was more strongly depleted for variation than the branch point (2.21 vs. 2.42 variants per 100 bp; Fisher's exact test: $p = 2.05 \times 10^{-5}$, Fig. 6, Supplementary Data 9). Both positions were depleted for variation more strongly in heptamers with "TNA" motifs ($n = 184,468$; 91.40%). The lower variability of the thymine residue at position 4 compared to the branch point adenine was more pronounced (1.87 vs. 2.22 variants per 100 bp; Fisher's exact test: $p = 2.23 \times 10^{-13}$) at the "TNA" than other motifs.

The Ti/Tv ratio was lower for the branch point (1.34) and position four (1.46) of the heptamers with a canonical "TNA" motif than other genomic features tested except the splice sites, and it was significantly lower (Fisher's exact test: $p < 3.9 \times 10^{-16}$)

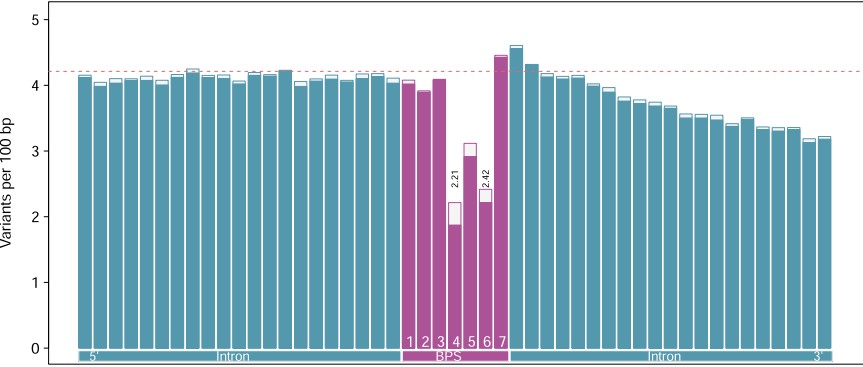

**Fig. 6 Local variation at the human branch point sequence (BPS).** The height of the bars indicates the variation (variants per 100 bp) surrounding 201,832 branch points while the solid bars indicate the variation surrounding a subset of 184,468 branch points encompassed by a heptamer with a canonical ("TNA") motif. The dotted horizontal red line indicates the average genome-wide variation.

compared to intronic adenine (1.97) and thymine (1.94) residues (Supplementary Note 1).

Branch point sequences in the human genome were also predicted with LaBranchoR[19] and branchpointer[20], and subsequently compared to the predictions from BPP (Supplementary Note 2). The pattern of variability was similar for the 102,944 introns where the BPS prediction matched across the three tools (Supplementary Fig. 6). As observed in the cattle data, the variability was lower for branch point sequences predicted by all three tools than only by BPP. The variability was significantly (Fisher's exact test: $p = 7.82 \times 10^{-5}$) lower at the conserved fourth position of the heptamer (1.82 variants per 100 bp) than the branch point (2.06 variants per 100 bp) in introns for which the prediction matched across tools. For a subset of 99,632 branch point sequences that matched across three tools, the heptamer contained a canonical "TNA" motif. In these heptamers, the variability at the branch point and thymine residue was respectively 2.01 and 1.64, suggesting a stronger constraint on the fourth residue of the heptamer than the branch point itself (Fisher's exact test: $p = 1.0 \times 10^{-9}$).

**Bovine BPS are enriched for splicing QTL.** To detect variants affecting splicing, we established a splicing QTL mapping cohort. We collected an average of 283.6 million reads from total RNA extracted from testis tissue of 76 bulls that also had their genomes sequenced at an average of 12.6-fold coverage. Using the GATK-based multi-sample DNA sequence variant genotyping, we called genotypes at polymorphic sites and retained 13,624,350 sequence variants that had a minor allele frequency greater than 0.05. RNA reads were aligned to the bovine reference genome using a splice-aware alignment tool. Intron excision ratios were subsequently inferred using the Leafcutter software and RegTools, and subsequently used as molecular phenotypes in a cis-splicing QTL analysis. In total, we detected 583,375 and 495,614 variants that were associated ($P_{adj} < 0.001$) with differential splicing based on exon junctions that were inferred using Leafcutter and RegTools, respectively. An exonic 58373887C > T variant (rs474302732) that was previously described to activate cryptic splicing in Brown Swiss cattle[24,25] was the top variant ($P_{adj} = 4.79 \times 10^{-20}$) at a cis-acting splicing QTL on bovine chromosome 6 (Supplementary Fig. 7), confirming that our mapping cohort enables us to readily reveal variants affecting molecular phenotypes.

In the sQTL mapping cohort, 4435 variants with minor allele frequency > 0.05 resided within the predicted heptamer encompassing the branch point, including 416 and 470 variants affecting the conserved fourth and sixth position, respectively. The seven bases upstream and downstream contained more variants (5743 and 6078) than the heptamer, corroborating that the BPS is

depleted for mutations. When compared to all variants located within 50 kb of molecular phenotypes, variants within the heptamer were enriched among putative cis-sQTL ($P_{Leafcutter} = 1.37 \times 10^{-54}$, $P_{RegTools} = 8.05 \times 10^{-43}$). However, variability within the seven bases up- and downstream the heptamer was not statistically different from the heptamer itself, suggesting that variants nearby the 3' splice site are generally enriched for splicing QTL. Widespread linkage disequilibrium (LD) may further lead to the presence of many significantly associated variants nearby sQTL, thus preventing us to readily separate true sQTL from variants in LD.

**Variants affecting bovine BPS are associated with alternative splicing events.** Variants affecting predicted branch point sequences were also top variants at sQTL, suggesting that they contribute to alternative splicing events. For instance, an A > G-polymorphism at BTA29:50695659 was the most significantly associated variant ($P_{Leafcutter} = 9.86 \times 10^{-19}$, $P_{RegTools} = 1.64 \times 10^{-18}$) for a differential splicing event between the first and second noncoding exon of the IRF7 gene encoding interferon regulatory factor 7 (Fig. 7, Supplementary Data 10 & 11). The BTA29:50695659 A allele was predicted with high confidence (BPP score: 7.30) as a branch point residue 47 bases upstream the start of IRF7 exon 2. LaBranchoR also predicted a branch point sequence encompassing the BTA29:50695659 A allele (position 7 of the heptamer). The predicted heptamer was outside the search space of branchpointer (44 nucleotides upstream of the 3' splice site), hence its prediction did not overlap with the other two tools.

The alternative G allele at BTA29:50695659 distorts the predicted branch point sequence and thereby introduces an AG dinucleotide. However, this AG dinucleotide is rarely used as 3' splice site. Instead, the BTA29:50695659 G allele is associated with utilization of an alternative canonical 3' splice acceptor site, leading to an alternative start of the second exon at 50,695,643 bp (Fig. 7e–g). This putative alternative exon start is supported by 27 heterozygous and 21 homozygous individuals, respectively, with an average of 28 and 49 reads spanning the splice junction. There was no support for an alternative splicing event detected in 27 individuals that were homozygous for the BTA29:50695659 A allele. While this putative alternative exon is missing in the Ensembl annotation (version 104) of the bovine genome, it is part of the Refseq (version 106) annotation for IRF7 (NM_001105040). Intriguingly, all three tools predicted a heptamer encompassing a canonical "TNA" motif 23 bases (position of the A allele) upstream of the alternative start site of IRF7 exon 2 at 50,695,620 bp (Fig. 7g).

We detected a C > G polymorphism (BTA29:43712184) affecting a "CNA" motif of a predicted branch point sequence

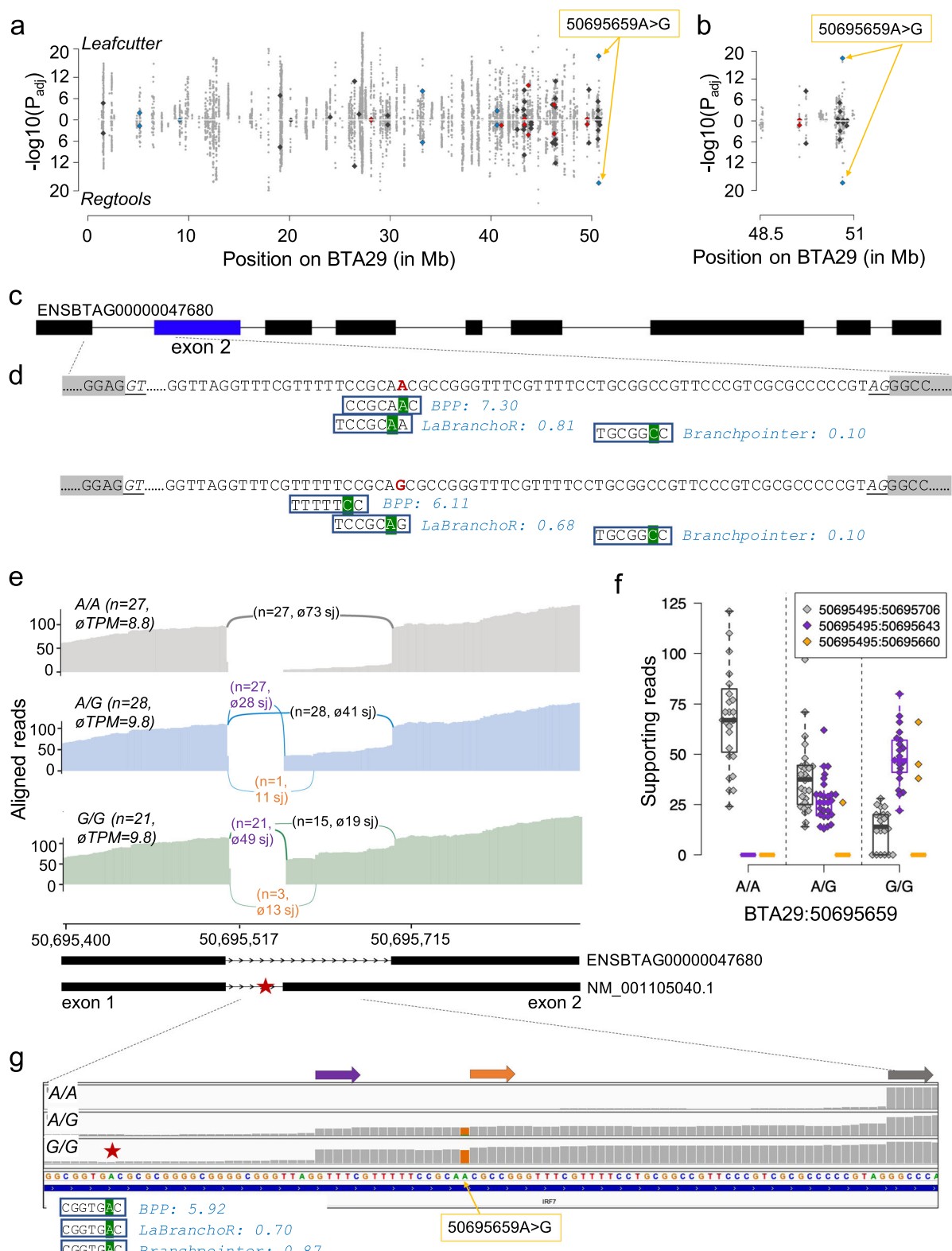

that was placed with high confidence 26 bases upstream the 3' splice site of *SCYL1* (*SCY1 like pseudokinase 1*) intron 9 by all three tools (BPP score: 7.16, LaBranchoR probability: 0.73, branchpointer probability = 0.68, Fig. 8, Supplementary Data 10 & 11). SCYL1 was expressed with $46 \pm 7$ TPM in testis transcriptomes of 76 bulls. While both BPP and LaBranchoR placed the branch point for the mutant G allele in different locations and with lower confidence, suggesting alternative

branch point usage, branchpointer predicted a branch point with a noncanonical "GNA" motif at the same location for the 43712184 G allele (albeit with slightly lower probability).

The BTA29:43712184-variant was also the top variant ($P_{\text{Leafcutter}} = 2.8 \times 10^{-10}$, $P_{\text{RegTools}} = 6.11 \times 10^{-5}$) at a cis-sQTL. The BTA29:43712184 G allele is associated with alternative 3' splicing 66 bases upstream the regular splice acceptor site of *SCYL1* exon 10. This putative alternative splice site is supported

**Fig. 7 A branch point mutation is associated with the utilization of alternative 3' splice acceptor sites in *IRF7*.** Manhattan plot of cis-acting splicing quantitative trait loci on chromosome 29. Intron excision ratios were calculated based on exon junctions that were extracted from RNA sequencing read alignments using either Leafcutter or RegTools. Orange arrows indicate the BTA29:50695659A > G variant. Red, blue, and dark-grey symbols represent variants affecting the fourth, sixth and all other positions of the heptamer, respectively (**a**). Detailed view of the region encompassing the BTA29:50695659A > G variant (**b**). Structure of the bovine *IRF7* gene (ENSBTAG00000047680). Boxes represent exons (**c**). Branch point sequences predicted in the first intron of *IRF7*. The BTA29:50695659 A allele (red colour) was the most likely branch point predicted by BPP, and it is located at the seventh position of a heptamer encompassing a "GNA" motif that was predicted as the most likely branch point by LaBranchoR. Prediction from branchpointer did not match the other two tools as this heptamer was outside its search space. Green background highlights the predicted branch point residue. Numbers reflect scores predicted by the different tools. The branch point was either placed at a different location (BPP) or predicted with less confidence (LaBranchoR) for the intronic sequence with the BTA29:50695659 G allele (**d**). Sashimi plots of RNA sequencing coverage and splice junction (sj) utilization in 76 animals of the sQTL cohort for different BTA29:50695659 genotypes (grey: A/A, blue: A/G, green: G/G). The abundance of *IRF7* mRNA in testis tissue is indicated in transcripts per million (TPM). Arcs indicate splice junction reads, with the thickness of the arc representing the average number of reads spanning the two exons. Values in parentheses indicate the number of samples (n) for each genotype as well as the average number of junction-spanning reads (**e**). Boxplots and beeswarm plots of the number of reads supporting the primary (grey) and two alternate (purple, orange) splice junctions (sj) (**f**). Representative IGV screenshots of the RNA sequencing alignments for three animals with different BTA29:50695659 genotypes. Grey, purple, and orange arrows represent alternative exon starts. The red star indicates a branch point sequence with a canonical "TNA" motif that was consistently and confidently predicted by all three tools upstream the alternative start of exon 2 at 50,695,643 bp (**g**).

by 18 heterozygous animals with an average of 86 reads and a single homozygous animal with an average of 211 reads spanning the exon junction. There was no support for an alternative splicing event in 57 animals that carried the wild type "CNA" motif in the homozygous state.

The in-frame addition of 66 coding nucleotides extends SCYL1 by 22 amino acids (EAPPMCGTLPTPAHTVRSPALQ)—adding them between the residues 410 and 411—thereby interrupting the second HEAT (Huntingtin, elongation factor 3 (EF3), protein phosphatase 2A (PP2A), yeast kinase TOR1 domain) repeat domain[26]. We did not detect this putative alternative protein variant in neither known paralogs nor orthologs of SCYL1 in 84 placental mammals that we investigated from the most recent Ensembl database (version 104). While alleles disrupting the expression of *SCYL1* may result in neurodegenerative diseases[26–28], the functional significance of the BTA29:43712184 G allele remains unknown. It is possible that a lower amount of wild type mRNA in animals with the BTA29:43712184 G allele is sufficient to maintain the physiological role of SYCL1. Since we investigated alternative *SYCL1* splicing only in testis transcriptomes, we cannot preclude that the alternative splicing event associated with the BTA29:43712184 variant is tissue-specific.

The BTA8:95322999A > C variant is another example for a sQTL that coincides with a predicted BPS. The BTA8:95322999 A allele resides within a putative BPS that was predicted by all three tools (Supplementary Fig. 8). According to the LaBranchoR and branchpointer prediction, the A allele constitutes the branch point in *FSD1L* (*fibronectin type III and SPRY domain containing 1 like*) intron 7. All three tools placed the branch point at different locations for the intronic sequence containing the BTA8:95322999 C allele. Interestingly, branchpointer and BPP predicted heptamers with canonical "TNA" motifs for the intronic sequence containing the nonreference C allele. The BTA8:95322999A > C variant is also the top variant at a splicing QTL that was associated with alternative 3' splice site utilization at *FSD1L* exon 8. However, most junctions detected in bulls carrying the C allele correspond to the regular splice site (Supplementary Fig. 8d, e), possibly due to utilization of an alternative—but nearby—branch point.

## Discussion

We used a catalogue of 29.4 million autosomal variants detected in 266 cattle to investigate evolutionary constraints on coding and noncoding features of the bovine genome. Our results show that the essential splice sites are less variable than any other feature investigated, suggesting extreme purifying selection. The strong constraint on splice regions of the bovine genome mirrors findings in human genomes[29,30]. While the splice acceptor and splice donor sites are accessible from gene annotation files, other motifs that are required for spliceosome assembly are barely annotated and thus poorly characterized. This motivated us to scrutinize the branch point sequence, a cis-acting intronic feature involved in pre-mRNA splicing. To the best of our knowledge, branch point sequences have neither been experimentally validated nor computationally predicted in bovine introns so far. Thus, our study uncovers a hitherto neglected feature of the bovine genome and investigates its evolutionary constraints at nucleotide resolution.

We considered up to 250 bp upstream of the 3' splice site to predict branch point sequences. A vast majority (81%) of the bovine branch points are between 18 and 37 bp upstream of the 3' splice site, corroborating previously reported constraints on the placement of branch points relative to the 3' splice site[9–11]. While all four nucleotides may act as branch points in principle, a preference toward adenine and cytosine over uracil and guanine was observed in early mutational experiments[31]. We observed a very strong preference towards adenine at both the bovine and human branch points. The preference towards adenine is stronger for the branch points detected in our study than for experimentally identified branch points in human introns[11,16]. This indicates that the computational prediction of branch point sequences as applied in our study prioritizes heptamers with canonical motifs.

The assembly of the spliceosome requires base pairing between a heptamer encompassing the branch point and the branch point recognition sequence "GUAGUA" of the U2 snRNA. Thus, the complementary sequence TACTAAC (the underlined branch point does not pair with the U2 snRNA, bulges out and acts as a nucleophile) was proposed as the most preferred branch site for mammalian pre-mRNA splicing[32]. Our findings show that the TACTAAC motif rarely (0.22%; $n = 393$) constitutes the branch point sequence in bovine introns. This is different from yeast, where the TACTAAC motif is deeply conserved[9,33]. The consensus motif of the branch point sequence in bovine introns includes conserved thymine and adenine residues at positions 4 and 6. This motif resembles branch point sequences detected in other mammalian species[9,10,16,34,35]. Moreover, the purine and pyrimidine content of the last five residues is similar for predicted branch point sequences in cattle, other mammals, and yeast. Thus, our findings in 179,476 introns suggest that the branch point in bovine introns is contained within a degenerate heptamer with a consensus sequence of "nnyTrAy."

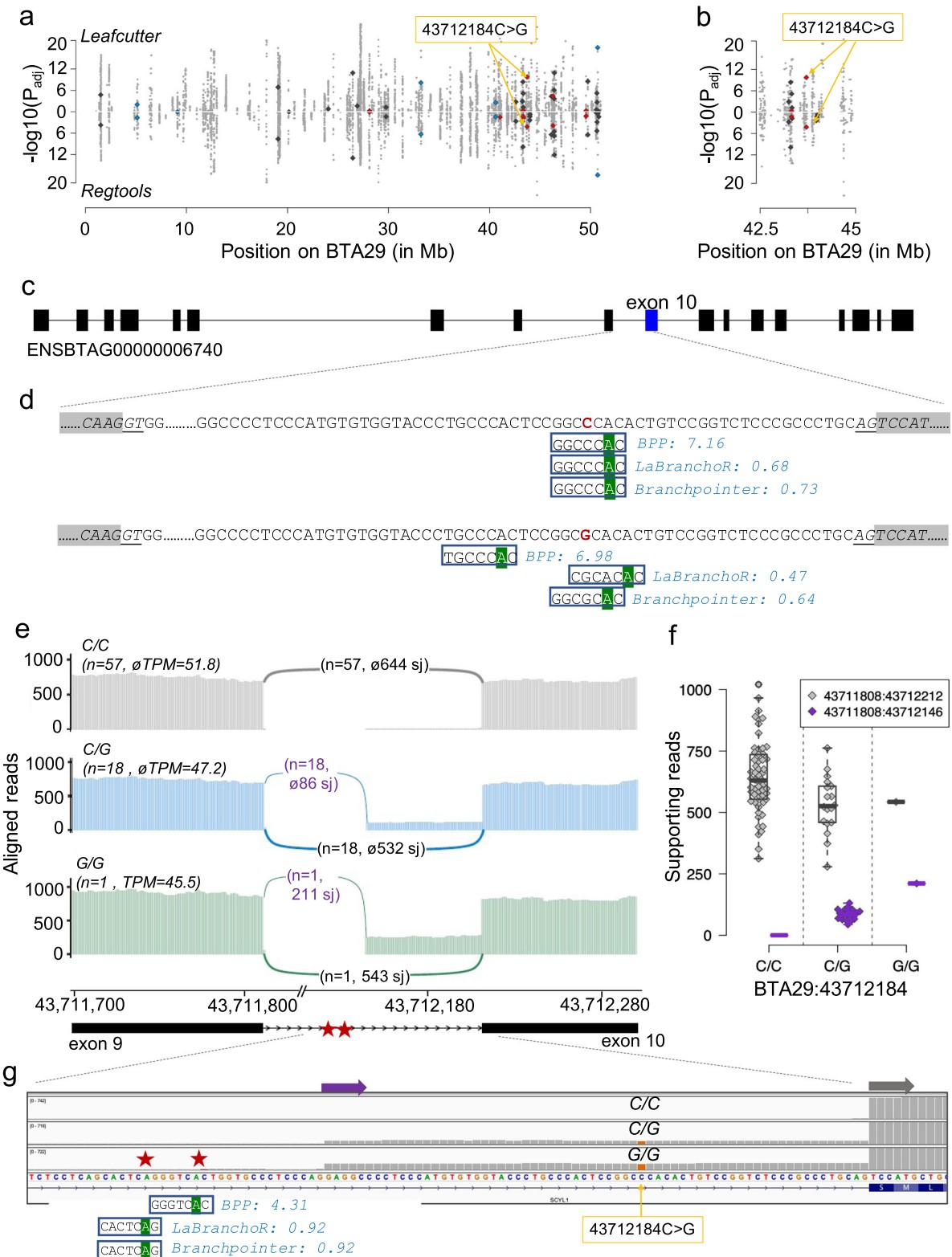

Counting the number of variable sites within functional classes of annotations is a straightforward approach to study evolutionary constraints on genomic features[29,36,37]. Using a variant catalogue established from the whole-genome sequencing of 266 cattle, we show that this method uncovers constraints at nucleotide resolution. For instance, we retrieved signals of purifying selection at the two essential splice sites, as well as known patterns within coding triplets. The confirmation of well-known

constraints suggests that this approach, despite its simplicity, reveals biologically valid signatures. The overall variability of the tested features is sensitive to the size of the variant catalogue, which in turn depends on the number of samples sequenced[29]. A larger variant catalogue in our human than cattle dataset yielded higher variation within all features studied. However, feature variability relative to the genomic average was similar in cattle and human, suggesting that constraints are comparable in both

**Fig. 8 A mutation within a predicted branch point sequence is associated with alternative 3′ splicing of *SCYL1* exon 10.** Manhattan plot of cis-acting splicing quantitative trait loci on chromosome 29. Intron excision ratios were calculated based on exon junctions that were extracted from RNA sequencing read alignments using either Leafcutter or RegTools. Orange arrows indicate the BTA29:43712184C > G variant. Red, blue, and dark-grey symbols represent variants affecting the fourth, sixth and all other positions of the heptamer, respectively (**a**). Detailed view of the region encompassing the BTA29:43712184C > G variant (**b**). Structure of the bovine SCYL1 gene (ENSBTAG00000006740). Boxes represent exons Boxes represent exons (**c**). Branch point sequences predicted in *SCYL1* intron 9. The BTA29:43712184 C allele (red colour) resides within a heptamer encompassing a "CNA" motif that was consistently and confidently placed at the same location by all three tools. Green background highlights the predicted branch point. Numbers reflect scores predicted by the different tools. The branch point was either placed at a different location (BPP, LaBranchoR) or predicted with less confidence (branchpointer) for the intronic sequence with the BTA29:43712184 G allele (**d**). Sashimi plots of RNA sequencing coverage and splice junction (sj) utilization in 76 animals of the sQTL cohort for different BTA29:43712184 genotypes (grey: C/C, blue: C/G, green: G/G). The abundance of *SYCL1* mRNA in testis tissue is indicated in transcripts per million (TPM). Arcs indicate splice junction reads, with the thickness of the arc representing the average number of reads spanning the two exons. Values in parentheses indicate the number of samples (*n*) for each genotype as well as the average number of junction-spanning reads (**e**). Boxplots and beeswarm plots of the number of reads supporting the primary (grey) and alternate (purple) splice junctions (sj) (**f**). Representative IGV screenshots of the RNA sequencing alignments for three animals with different BTA29:43712184 genotypes. Grey and purple arrows represent alternative exon starts. Red stars indicate putative branch point sequences predicted upstream the alternative start of exon 10 at 43,712,146 bp (**g**).

species. It is however worth noting that the large catalogue of variants in the human dataset enabled us to better distinguish variability in individual residues.

The variability at positions 4 and 6 of the predicted branch point sequences was strikingly lower than at surrounding intronic nucleotides and at coding sequences including the highly constrained second residue of coding triplets both in the bovine and human dataset. These two intronic residues are strongly conserved in branch point sequences across vertebrates[11,38]. The fact that two evolutionarily conserved residues within the heptamer are strongly depleted for variants, suggests that they are under extreme purifying selection. This finding also suggests, despite the lack of experimental evidence, that the branch point sequence constitutes an evolutionarily constrained regulatory element in the bovine genome. Our splicing QTL analysis showed that these intronic thymine and adenine residues harbor variants that are associated with alternative pre-mRNA splicing. Variants at these residues that perturb splicing have been associated with diseases in humans[10,13,14,39,40]. However, to the best of our knowledge, phenotypic consequences of mutated branch point sequences have not been described yet in cattle. This is surprising given our observations of the strong constraint on the predicted branch point sequence and their potential impacts on mRNA splicing. We suspect that this is partly because disease-causing variants are often assumed to reside in coding sequences. Intronic variants that are distant to the readily annotated splice acceptor and donor sites are widely neglected as candidate causal variants[41–46], likely due to the poor annotation of the noncoding features in the bovine genome[47]. Our study provides evidence that the bovine branch point sequence constitutes a feature that harbors alleles with pronounced impacts on pre-mRNA splicing. We provide the coordinates of the predicted branch point sequences to make them readily amenable to future genetic investigations (Supplementary Data 1). For instance, the genotyping of variants affecting predicted branch point sequences using customized genotyping arrays in large cohorts might enable differentiating between the tolerated and deleterious variants affecting this hitherto neglected intronic feature[48].

The thymine residue at the fourth position of the heptamer is conserved across vertebrates more strongly than the branch point adenine itself[11]. Our study revealed that the thymine residue is strongly depleted for mutations in both cattle and humans. While the evolutionary constraint was similar on the thymine and adenine residue in predicted cattle branch point sequences, the constraint was stronger on the thymine than branch point residue in human branch point sequences. Its intolerance to mutations is likely due to the fact that the conserved thymine residue is the

only nucleotide in the heptamer that cannot undergo wobble base pairing with the recognition site of the U2 SnRNA[11]. Moreover, it has a role (along with the branch point) in binding to the conserved pre-mRNA splicing factor SF1 in the early steps of spliceosome assembly[49]. In contrast, nucleotides other than adenine may serve as branch points[31]. We observed an excess of transversions among the SNPs affecting the thymine and adenine residues within the branch point sequence. Considering that cytosine is the second most preferred branch point residue, it seems plausible that some SNPs substituting the branch point adenine by cytosine do not compromise branch point activity. Disease-associated mutations affecting the two highly conserved residues at human branch point sequences are enriched for transitions[16,40,50]. Thus, the excess transversions observed at the two nucleotides suggest that the variants detected in our cohort of healthy individuals are possibly enriched for tolerated variations.

The computationally predicted branch point sequences of our study likely contain falsely placed branch points. We suspect that true branch points are even more strongly depleted for variation than suggested by our study. The fact that we observed less variation in a subset of branch point sequences that were predicted by three methods supports this hypothesis. On the other hand, our study considered only the most likely branch point per intron. Introns with multiple branch points are prevalent in the human genome[11] and their branch points might be more tolerant to mutations due to the availability of alternative branch points. In any case, our findings indicate that systematic assessment of sequence variation affecting branch point sequences is warranted.

## Methods

**The bovine variant catalogue.** We considered single-nucleotide polymorphisms (SNPs) identified in whole-genome sequence data of 266 cattle from 11 breeds to investigate variation in various bovine genomic features. All animals were sequenced with Illumina instruments using short paired-end reads at an average coverage of 10-fold or higher (mean = 16.3). Accession numbers for the raw sequencing data of all animals of the present study are listed in Supplementary Data 2.

Following removal of adapter sequences using the fastp software package[51], high-quality reads were aligned to the ARS-UCD1.2 assembly of the bovine genome[21] using the mem-algorithm of the BWA software[52]. We marked duplicate reads using Picard tools (http://broadinstitute.github.io/picard/) and sorted the alignments by coordinates using Sambamba[53]. The base quality scores were recalibrated using the BaseRecalibrator module[54] of the Genome Analysis Toolkit[55] (GATK - version 4.1.4.1) using 110,270,189 unique positions from the bovine dbSNP version 150 as known variants.

Variant discovery and genotyping were carried out on the 29 autosomes following the GATK best practice guidelines (https://gatk.broadinstitute.org/hc/en-us/articles/360035535932-Germline-short-variant-discovery-SNPs-Indels-). We applied the HaplotypeCaller, GenomicsDBImport and GenotypeGVCFs modules from the GATK (version 4.1.4.1)[56] to discover and genotype SNPs and INDELs.

Site-level hard filtering was applied using the VariantFiltration module of GATK and the raw genotypes were refined, and sporadically missing genotypes were imputed using beagle4.1[57]. The 29,418,150 high-quality SNPs (29,227,950 biallelic and 190,200 multiallelic) passing the hard filtering and segregating in 266 samples were subsequently used to study variation in bovine genomic features both at feature level and at nucleotide resolution. The genomic relationship matrix among the sequenced animals was constructed using 20,553,068 genomic variants segregating with a minor allele frequency >0.01 and principal component analysis was performed using GCTA[58].

**Estimation of variation in functional features**. We obtained the coordinates of genomic features from the Ensembl (release 104) annotation of the bovine reference genome[21]. We considered 35,848 transcripts from 20,785 protein-coding genes on the 29 autosomes with 218,245 exons and 190,203 introns, as well as 17,759 3' UTR, 25,490 5' UTR, 24,217 stop codons 23,583 start codons, 190,203 3' splice sites, and 190,203 5' splice sites. Each nucleotide from the bovine reference genome (2,489,368,272 nonmissing autosomal nucleotides) was assigned to one of the eight functional features according to the following priority order: stop codon, start codon, 5' splice site, 3' splice site, 3' UTR, 5' UTR, exons, introns. Bases ($n = 2,010,448,274$) that did not overlap any of the eight features or the 2090 lncRNAs were assigned to the intergenic category. Using the previously established variant catalogue (see above), we assessed the variability of each feature as the number of SNPs per 100 nucleotides assigned to the feature. The allele frequency distribution within each feature was assessed based on the minor allele frequency of all variants overlapping the features.

Similarly, the variation at the nucleotide level was estimated by considering one nucleotide of a feature at a time. For instance, to estimate the variation at the first residue of the 5' splice site, we extracted the corresponding coordinates from $n = 190,203$ 5' splice sites. Thus, the variation at the first residue of the 5' splice site was expressed as $100 \times \frac{m}{n}$ (variants per 100 bp), where m was the number of SNPs overlapping these coordinates.

**Prediction of branch point sequences**. We used a mixture model implemented in the software program BPP[18] to predict branch point sequences in the bovine introns. First, we obtained the weighted octanucleotides from the bovine introns following the approach proposed by Zhang and colleagues[18]. Briefly, we extracted 14 bp long intronic sequences from the polypyrimidine tract region (3–16 bp upstream of the 3' splice site) and the background region (187–200 bp upstream of the 3' splice site) of 146,992 introns that were longer than 300 bp. The octanucleotide frequency was calculated in both regions separately and scored as proposed by Zhang and colleagues[18].

Next, using these weighted octanucleotides and the position weight matrix (PWM) of predicted human branch point sequences (downloaded from https://github.com/zhqingit/BPP), we predicted the branch point sequence in bovine introns that were at least 20 bp long. We considered 179,476 unique intronic sequences (up to 250 bp upstream of 3' splice site) from protein-coding transcripts. Within each intron, we retained the branch point sequence with the highest score.

Branch point sequences were also predicted using two other tools, LaBranchoR[19] and branchpointer[20], with default parameters. LaBranchoR restricts the search for branch point sequences to introns that are at least 70 bases long. So, we considered 177,668 introns from protein-coding transcripts that were at least 70 bp long for branch point prediction with LaBranchoR. The search space of branchpointer is limited to the intronic sequence between 18 and 44 bp upstream of the 3' splice site. So, 178,559 introns that were longer than 44 bp were considered for branch point sequence prediction with branchpointer. Only the branch point sequence with the highest score was retained from all tools for further analyses.

**Validation with human data**. The coordinates for functional features were obtained from the gene transfer file of the Ensembl annotation (release 104) of the human genome (GRCh38). The human branch point sequences were predicted in 201,832 introns (>20 bp long) using the software program BPP[18]. The weighted octa-nucleotides and PWM were downloaded from https://github.com/zhqingit/BPP. Branch points in the human genome were also predicted using LaBranchoR[19] and branchpointer[20] for introns longer than 69 and 44 bp, respectively (see above).

A variant catalogue was established from publicly available human data. Specifically, we downloaded vcf-files from the Human Genome Diversity Project (HGDP) and the 1000 genomes project (1KG) from the genome aggregate database (gnomAD) v3.0 (https://gnomad.broadinstitute.org/blog/2019-10-gnomad-v3-0/). The vcf files contained genotypes for 115,640,370 high-quality SNPs (passing the VQSR) segregating in 3942 samples of diverse ancestry.

**Establishing a splicing QTL mapping cohort**. Testis tissue of 76 mature bulls was collected from a commercial slaughterhouse after regular slaughter. Samples were snap-frozen within 5 h postmortem and stored at −80 °C. Total RNA and DNA was extracted from tissue samples using AllPrep DNA/RNA Mini Kits (Qiagen), according to the manufacturer's instructions. The integrity and concentration of RNA and DNA were analyzed by agarose gel electrophoresis and

Qubit 2.0 fluorometer (ThermoFisher), respectively. The RNA integrity number (RIN) was assessed with the Bioanalyzer RNA 600 Nano assay (Agilent Technologies).

DNA samples were sequenced on an Illumina NovaSeq6000 using 150 bp paired-end sequencing libraries. Quality control (removal of adapter sequences and bases with low quality, and trimming of poly-G tails) of the raw sequencing data was carried out using the fastp software (version 0.19.4)[51] with default parameters. Following quality control, between 70,493,763 and 307,416,205 read pairs per sample were aligned to the ARS-UCD1.2 version of the bovine reference genome[21] using the mem-algorithm of the BWA software (version 0.7.17)[52] with option -M to mark shorter split hits as secondary alignments. Sambamba[53] (version 0.6.6) was used for coordinate-sorting and to combine read group-specific BAM files into sample-specific sorted BAM files. Duplicated reads of the merged and coordinate-sorted BAM files were marked using the MarkDuplicates module from Picard tools (https://broadinstitute.github.io/picard/). The mosdepth software[59] (version 0.2.2) was used to extract the number of reads that covered a genomic position in order to obtain the average coverage per sample and chromosome. We considered only high-quality reads (by excluding reads with mapping quality <10 and SAM flag 1796) to obtain the average coverage. The average coverage of the 76 samples ranged from 6.3 to 27.6 with a mean value of 12.6 ± 4.2. Sequence variant genotypes were called using the multi-sample variant genotyping approach implemented with the HaplotypeCaller, GenomicsDBImport and GenotypeGVCFs modules from GATK as described above. Site-level hard-filtration was applied using the VariantFiltration module of GATK and the raw genotypes were refined using Beagle as described above. We considered 13,624,350 sequence variants with a minor allele frequency greater than 0.05 to detect splicing QTL.

RNA sequencing libraries (2 × 150 bp) were prepared using the Illumina TruSeq Stranded Total RNA sequencing kit and sequenced on an Illumina NovaSeq6000. Quality control (removal of adapter sequences and bases with low quality, and trimming of poly-A and poly-G tails) of the raw sequencing data was carried out using the fastp software (version 0.19.4)[51] Following quality control, between 191,160,837 and 386,773,085 filtered reads per sample (mean: 283,587,831 ± 43,284,185) were aligned to the ARS-UCD1.2 reference sequence and the Ensembl gene annotation (release 104) using the splice-aware read alignment tool STAR (version 2.7.9a)[60] with options–twopassMode Basic,–sjdbOverhang 100,–outFilterMismatchNmax 3, and–outSAMmapqUnique 60. Transcript abundance (in transcripts per million, TPM) was quantified using kallisto[61] and aggregated to the gene level using the R package tximport[62].

**Detection of splicing QTL**. In order to quantify RNA splicing variation among the sequenced samples, we used the Leafcutter software[63]. First, we created the junction files from the STAR-aligned bam files (see above) using either the bam2junc.sh utility from Leafcutter or the junctions extract utility from RegTools[64]. Next, we performed intron clustering, calculated intron excision ratios, and extracted the five principal components of the sample covariance matrix using the leafcutter_cluster.py and prepare_phenotype_table.py utilities, respectively. Subsequently, we used FastQTL[65] to detect cis-splicing QTL (cis-sQTL) within a cis-window of 50,000 bp of molecular phenotypes while fitting the top five principal components as covariates. Association between molecular phenotypes and variants with minor allele frequency >0.05 was tested with the identified splice junctions. Specifically, for splice junctions determined via Leafcutter, 522,348,087 tests were carried out with 1,050,381 molecular phenotypes and 10,856,228 variants, and for splice junctions determined via RegTools 149,659,456 tests were conducted with 305,114 molecular phenotypes and 9,665,065 variants. To account for multiple testing, we applied Bonferroni correction and considered variants that had adjusted P-values less than $1 \times 10^{-3}$ as putative cis-sQTL. Putative cis-sQTL were visualized using ggsashimi[66]. We applied Fisher's exact tests to investigate the overlap between cis-sQTL and variants affecting predicted branch point sequences.

**Reporting Summary**. Further information on research design is available in the Nature Research Reporting Summary linked to this article.

## Data availability

The raw sequences used to establish the variant catalogue and the splicing QTL mapping cohort are available from the European Nucleotide Archive with the accession numbers listed in Supplementary Data 2 and 3, respectively. The human variant catalogue is available via gnomAD (https://gnomad.broadinstitute.org/blog/2019-10-gnomad-v3-0/).

## Code availability

Processing of the raw sequencing data and prediction of branch point sequence were carried out using public software as indicated in the material and methods section. Statistical analyses and graphical visualizations were performed using python version 3.8.3 and R version 3.5.1.

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

## Acknowledgements

This study was supported by grants from the Swiss National Science Foundation (310030 185229), the European Union's Horizon 2020 research and innovation programme under Grant Agreement No. 815668 (BovReg), an ETH Research Grant, and the Swiss Federal Office for Agriculture. The funding bodies were neither involved in the design of the study and collection, analysis, and interpretation of data nor in writing the manuscript. We thank Maya Hiltpold for support in sampling testis tissue and the Functional Genomics Center Zurich for generating DNA and RNA sequencing data.

## Author contributions

N.K.K. and H.P. conceived and designed the experiments and analysed data. X.M.M. sampled tissue and prepared DNA and RNA for the sQTL cohort. N.K.K. and H.P. wrote the paper. All authors read and approved the submitted manuscript.

## Competing interests

The authors declare no competing interests.
