## [Transparent Peer Review File · Communications Biology]

Reviewers' comments:

Reviewer #1 (Remarks to the Author):

Summary:

In this work Kadri *et al.*, present the evolutionary constraints of the branch point motifs within the bovine genome. The authors sequenced the genome of 266 cattle and identified 29.4 millions of variants. By computational methods, BPP and then LaBranchoR, a comprehensive list of putative branch point motifs was determined for the bovine genome. Then the authors described the repartition of variants along the branch point motif to assess the conservation of this motif. The results obtained on bovine branch points were compared to the results of human branch points by the same method. The authors applied this strategy on the donor and acceptor consensus splice site motifs to confirm that this strategy can detect the conservation of the well-known acceptor and donor canonical motifs (AG/GT).

Overall, we appreciate the effort to study the branch point motif in bovine genome. The importance of these short and degenerated motifs is often under-estimated, in particular for the molecular diagnosis. This work highlights the lack of knowledge regarding the annotation of non-human genome. We also acknowledge the important work done to gathering the sequencing data a large cohort of cattle ($n = 266$), with the constitution of large variant collection ($n = 29.4$ million of variants). Beyond the study of branch point motif characteristics, this variant collection should be useful for other genomic projects.

We have few suggestions that could further improve the manuscript and increase its scientific impact.

Major point:

The lack of experimentally proven branch point is the major limit of this work. Although we understand the difficulty to get a collection of experimentally proven branch points, the use of prediction tool, in particular BPP, could lead to several bias.

First, the most of prediction tool was trained on human intronic sequences. While splicing patterns are thought to be conserved across eukaryotes, to our knowledge there is no evidence that these predictive tools are as relevant in non-human species as in human.

Second, for BPP, the authors had used a set of putative branch point based on the sequence conservation as training set of BPP (Zhang *et al.*, 2017). So we think that BPP is not the optimal tool to study the conservation of branch point motif, because the study of BPP-predicted branch points could over-estimate the conservation of branch point motif. Thus the use of prediction tools trained on experimentally proven branch point, such as Branchpointer (Signal *et al.*, 2018), LaBranchoR (Paggi *et al.*, 2018) or RNABPS (Nazari *et al.*, 2019), seems to be the most relevant for this work.

Minor points:

- It could be interesting to study a possible correlation between the score value of branch point given by the tools and the presence of variant within the branch point motif.
- In parallel with the studying of branch point motif conservation, the research of variants introducing a novel AG di-nucleotide in the AG-exclusion zone between authentic 3'ss AG and the branch point (Wimmer *et al.*, 2020) could also confirmed the constraints on intronic sequence conservation.
- The authors claimed that the branch point motif is an heptamer, actually there is not a clear consensus on the real size of branch point motif, for example Gao *et al.* declared that the branch point motif is an pentamer, yUnAy (Gao *et al.*, 2008).
- Several bibliographic references date from before 2000, in particular for the definition of splicing motifs, an update of these references would be preferable.
- The authors claimed that the branch points are typically between 18 and 37 bases upstream of acceptor splice site, actually the branch point region is larger, between 18 and 44 bases upstream acceptor splice site (Mercer *et al.*, 2015).
- On the figure 1A, use a color code to identified the breeds

- The author compared the variability of branch point motif with the 42 surrounding nucleotides, but the 21 nucleotide downstream of branch point overlap another conserved motif, the polypyrimidine tract. This could introduce a bias in the comparison of variability of branch point motif with the rest of intronic sequence.
- In the discussion, it could be added that the presence of variant in the canonical motif "TNA" is partially explained by the redundancy of branch point motif in intronic sequence (Mercer *et al.*, 2015).

Reviewer #2 (Remarks to the Author):

In this manuscript, Kumar and co-workers compared evolutionary constraints in different genomic features, with a focus on intronic branch points, in bovine and human genomes using whole-genome sequencing data. Taken together, the features described in this study can be used to explore branchpoint sequence variability in these species. Some comments:

- 1- In this study, authors considered 35,848 transcripts from 20,785 bovine protein-coding genes (lines 424-425) and considered all SNPs/bases not overlapped with stop codon, start codon, 5' splice site, 3' splice site, 3' UTR, 5' UTR, exons, and introns of these transcripts as "intragenic" (lines 431-432). In fact, this is not the correct definition of intragenic SNPs/bases as many of them could overlap (exonic and/or intronic) with lincRNAs. For example, there are 2,199 lincRNAs in the current Ensembl bovine gene-build which are mostly spliced (96%). This issue is aggravating in the human genome which has around 14,000 annotated lincRNAs (Ensembl gene-build). Considering bases/SNPs overlapped with lincRNAs will likely bias lines 106-108 inference.
- 2- Figure 1B and lines 432-440. It's not clear how authors managed overlapped features (e.g., introns) to get the number of SNPs per 100 nucleotides assigned to each feature.
- 3- Lines 108-110, "The splice sites were the least variable feature with 0.12 and 0.17 variants per 100 bp, respectively, at 3' and 5' ends", please mention if these difference were statistically significant or not
- 4- Line 264, "MAF" needs to be changed to Minor Allele Frequency (MAF).

Reviewer #3 (Remarks to the Author):

I enjoyed reading the manuscript entitled "Quantifying evolutionary constraints on the intronic branch point sequence in the bovine and human genome".

Overall this manuscript is well and clearly written.

However, I do have a few comment for the authors to consider. The first is, did the authors consider different strandedness in these data sets. Second, is the inferences that the author made in lines 353-362. While some researchers (both cattle and human) still conform the the central dogma of coding variants being likely the candidates of disease, I am not convince that this is the majority. I think that the authors are overlooking some very important difference between the genetic tools and the data and phenotypic resources that are available to researchers of these two different species. The amount of data and the resolution of genotypic and phenotypic information that these two research communities have are vastly different. For example the largest genotypic array for cattle is approximately 700K SNPs as opposed to human arrays of 2.5M SNPs. Therefore, it is also possible that fewer disease causing splice variants have been describe in cattle because of lower genotypic resolution and decreased phenotypic reporting.

regarding line 353 -consider adding the word potential before fatal since this paper did not demonstrate this.

Finally, while BioRxiv is a great resource, perhaps the authors should site the final peer reviewed articles when possible.

Reviewer #1 (Remarks to the Author):

Summary:

In this work Kadri *et al.*, present the evolutionary constraints of the branch point motifs within the bovine genome. The authors sequenced the genome of 266 cattle and identified 29.4 millions of variants. By computational methods, BPP and then LaBranchoR, a comprehensive list of putative branch point motifs was determined for the bovine genome. Then the authors described the repartition of variants along the branch point motif to assess the conservation of this motif. The results obtained on bovine branch points were compared to the results of human branch points by the same method. The authors applied this strategy on the donor and acceptor consensus splice site motifs to confirm that this strategy can detect the conservation of the well-known acceptor and donor canonical motifs (AG/GT).

Overall, we appreciate the effort to study the branch point motif in bovine genome. The importance of these short and degenerated motifs is often under-estimated, in particular for the molecular diagnosis. This work highlights the lack of knowledge regarding the annotation of non-human genome. We also acknowledge the important work done to gathering the sequencing data a large cohort of cattle (n = 266), with the constitution of large variant collection (n = 29.4 million of variants). Beyond the study of branch point motif characteristics, this variant collection should be useful for other genomic projects.

We have few suggestions that could further improve the manuscript and increase its scientific impact.

Major point:

The lack of experimentally proven branch point is the major limit of this work. Although we understand the difficulty to get a collection of experimentally proven branch points, the use of prediction tool, in particular BPP, could lead to several bias.

First, the most of prediction tool was trained on human intronic sequences. While splicing patterns are thought to be conserved across eukaryotes, to our knowledge there is no evidence that these predictive tools are as relevant in non-human species as in human.

Second, for BPP, the authors had used a set of putative branch point based on the

sequence conservation as training set of BPP (Zhang *et al.*, 2017). So we think that BPP is not the optimal tool to study the conservation of branch point motif, because the study of BPP-predicted branch points could over-estimate the conservation of branch point motif. Thus, the use of prediction tools trained on experimentally proven branch point, such as Branchpointer (Signal *et al.*, 2018), LaBranchoR (Paggi *et al.*, 2018) or RNABPS (Nazari *et al.*, 2019), seems to be the most relevant for this work.

We thank the reviewer for their time to assess our manuscript and raising important concerns regarding the relevance of predictive tools that were trained on human data for predictions in cattle. To take these two concerns into account, we modified the manuscript at several places.

- First, we emphasize that pre-mRNA splicing is highly conserved across eukaryotes and that experimentally proven branch points are not available in cattle (lines 71-74, 481-482).
- Second, we have now predicted bovine and human branch point sequences with Branchpointer (Signal *et al.*, 2018) in addition to LaBranchoR (Paggi *et al.*, 2018) and BPP. We uncover similar patterns in local variability in the predicted heptamers from the three tools considered. Importantly, the overlap in predictions from the three tools was similar between the two species, which might indicate that they perform reasonably well for both humans and cattle. Notably, the constraint pattern of the heptamer was replicated across the three tools and both species and a stronger constraint was observed for predictions that matched across the three tools.

We provide a synthesis from the comparison in the main body of the revised manuscript and explain the results of our additional analyses in detail in Supplementary Note 2 including 4 novel Supporting Figures.

- Third, we performed a genome-wide splicing QTL study using testis transcriptomes and whole-genome sequencing data of 76 bulls. To the best of our knowledge, this dataset is the first of its kind investigated in cattle. We provide evidence that the computationally predicted branch points are enriched for splicing QTL. Moreover, we provide three examples of mutations affecting evolutionarily conserved residues of predicted branch points that lead to alternative splicing. These data are presented in the main body of the manuscript

(lines 28-29, 79-80, 320-444, 534-535 Figures 7 & 8) as well as in the supplement.

In conclusion, we think that these actions clarified the scope of our work and provide additional evidence that computationally predicted branch points are indeed evolutionarily conserved, and functionally relevant in cattle.

Minor points:

- It could be interesting to study a possible correlation between the score value of branch point given by the tools and the presence of variant within the branch point motif.

We thank the reviewer for this suggestion. We have now calculated the correlation between the prediction scores from three prediction tools and the variability of the predicted branch point. The results are described in brief in the results section in the main manuscript (line 216 onwards) and in detail in supplementary note 3.

- In parallel with the studying of branch point motif conservation, the research of variants introducing a novel AG di-nucleotide in the AG-exclusion zone between authentic 3'ss AG and the branch point (Wimmer *et al.*, 2020) could also confirmed the constraints on intronic sequence conservation.

We thank the reviewer for this suggestion. We have studied the frequency of mutations introducing novel AG di-nucleotides and describe it in a paragraph from line 228 onwards in the main text. A supplementary note (Supplementary note 2) with details of the analyses is included in this submission.

- The authors claimed that the branch point motif is an heptamer, actually there is not a clear consensus on the real size of branch point motif, for example Gao et al. declared that the branch point motif is an pentamer, yUnAy (Gao *et al.*, 2008).

We thank the reviewer for this comment. In fact, we don't intend to claim that the branch point motif is an heptamer. We strictly rely on previous literature to infer the corresponding statements. In order to avoid confusion, we modified statements that we thought were ambiguous accordingly (line 24-25 in abstract). For instance, we refer to Taggart et al., 2007, who showed that the branch point recognition sequence AUGAUG predominantly binds to a heptamer with a bulged branch point at position 6 in yeast.

These authors also find evidence for a similar model in humans. Since the branch point site recognition site of the U2snRNA is highly conserved across eukaryotes, including cattle, we hypothesize similar model is active in cattle.

Moreover, the branch point prediction tool BPP outputs heptamers as branchpoint sequence. To conform with the cited literature, and for simplicity we retained the heptamers as the branch point sequence from the prediction program, and refer to it as heptamers throughout the manuscript

- Several bibliographic references date from before 2000, in particular for the definition of splicing motifs, an update of these references would be preferable.

We respectfully disagree with this request, as we have attempted to refer to the primary sources throughout the manuscript.

- The authors claimed that the branch points are typically between 18 and 37 bases upstream of acceptor splice site, actually the branch point region is larger, between 18 and 44 bases upstream acceptor splice site (Mercer *et al.*, 2015).

We thank the reviewer for this comment. In fact, our results agree very well with Mercer *et al.*, 2015, who reported that 90% of human branch point sequences were between 19-37 bp upstream acceptor sites. In cattle, we find 81% to reside within 19 and 37 bp upstream acceptor sites. This has now been clarified in the main body of the manuscript (line 486-487).

- On the figure 1A, use a color code to identify the breeds

Figure 1A was indeed plotted with colours to separate animals by breeds. However, this colour code was inadvertently lost in the pdf production. We will include original pdf files of the plots with this submission.

- The author compared the variability of branch point motif with the 42 surrounding nucleotides, but the 21 nucleotide downstream of branch point overlap another conserved motif, the polypyrimidine tract. This could introduce a bias in the comparison of variability of branch point motif with the rest of intronic sequence.

We agree with the reviewer. This possible bias was already discussed in line 129 of the original manuscript when describing the local variation around splice sites (Figure 2).

In order to make this potential source of bias even clearer, we now mention the polypyrimidine tract while describing the local variation within and around the branch point sequence (Figure 4; line 199).

We however are confident that this does not affect our conclusions regarding the evolutionary constraint and local variability within the heptamer, as we compare the variability within branch point sequence to the mean variability of the coding sequence (and not the in the intron).

- In the discussion, it could be added that the presence of variant in the canonical motif “TNA” is partially explained by the redundancy of branch point motif in intronic sequence (Mercer *et al.*, 2015).

The redundancy (presence of multiple branch points) was already discussed in the earlier version. The corresponding statement can be found in line 569 of the revised manuscript.

Reviewer #2 (Remarks to the Author):

In this manuscript, Kumar and co-workers compared evolutionary constraints in different genomic features, with a focus on intronic branch points, in bovine and human genomes using whole-genome sequencing data. Taken together, the features described in this study can be used to explore branchpoint sequence variability in these species. Some comments:

1- In this study, authors considered 35,848 transcripts from 20,785 bovine protein-coding genes (lines 424-425) and considered all SNPs/bases not overlapped with stop codon, start codon, 5' splice site, 3' splice site, 3' UTR, 5' UTR, exons, and introns of these transcripts as "intragenic" (lines 431-432). In fact, this is not the correct definition of intragenic SNPs/bases as many of them could overlap (exonic and/or intronic) with lincRNAs. For example, there are 2,199 lincRNAs in the current Ensembl bovine gene-build which are mostly spliced (96%). This issue is aggravating in the human genome which has around 14,000 annotated lincRNAs (Ensembl gene-build). Considering bases/SNPs overlapped with lincRNAs will likely bias lines 106-108 inference.

We thank the reviewer for pointing this out. We have now considered 32,261,721 bases overlapping 2090 lincRNAs while assigning bases to intergenic regions of the genome. The estimate of variability of intergenic region increased slightly. (The number of variants per 100 bp is now 1.21 compared to the previous estimate of 1.12.) Figure 1 has been modified accordingly

Similarly, 488,127,421 bases overlapping 45,545 lincRNAs were considered for studying variability in human genomic features.

These modifications are mentioned in line 614 of the revised manuscript and in supplementary Note 1.

2- Figure 1B and lines 432-440. It's not clear how authors managed overlapped features (e.g., introns) to get the number of SNPs per 100 nucleotides assigned to each feature. Nucleotides between consecutive exons were assigned to the feature "introns". In case of alternatively spliced transcripts, if an intron is not spliced out and is retained as exon, by priority as explained in the methods (line 614) it will be considered exonic. So essentially a nucleotide can overlap only one feature in the order of defined priority.

3- Lines 108-110, “The splice sites were the least variable feature with 0.12 and 0.17 variants per 100 bp, respectively, at 3’ and 5’ ends”, please mention if these differences were statistically significant or not

Difference in variability of all possible pairs of bovine features is significant except between start codon and stop codon. This information has been added to the legends of Figure 1 in the main text, and of Figure 1 in Supplementary note 1.

4- Line 264, “MAF” needs to be changed to Minor Allele Frequency (MAF).

“MAF” has been changed to “minor allele frequency” throughout the revised manuscript.

Reviewer #3 (Remarks to the Author):

I enjoyed reading the manuscript entitled "Quantifying evolutionary constraints on the intronic branch point sequence in the bovine and human genome".

Overall this manuscript is well and clearly written.

However, I do have a few comment for the authors to consider. The first is, did the authors consider different strandedness in these data sets.

Yes, the different stranded-ness in the data was considered. To confirm that this was done correctly, we show that the variability pattern is consistent around branch point sequences on the positive (n=92,865, upper panel) and negative (n=86,611, lower panel) strand in the following plot. Please note that we provide this Figure only in the rebuttal letter and not in the main manuscript files.

Second, is the inferences that the author made in lines 353-362. While some researchers (both cattle and human) still conform the the central dogma of coding variants being likely the candidates of disease, I am not convicence that this is the majority. I think that the authors are overlooking some very important difference between the genetic tools

and the data and phenotypic resources that are available to researchers of these two different species. The amount of data and the resolution of genotypic and phenotypic information that these two research communities have are vastly different. For example the largest genotypic array for cattle is approximately 700K SNPs as opposed to human arrays of 2.5M SNPs. Therefore, it is also possible that fewer disease causing splice variants have been describe in cattle because of lower genotypic resolution and decreased phenotypic reporting.

We thank the reviewer for raising an important point. We agree that causal variants are not restricted to coding regions, and it was not our intention to make such a claim.

While the phenotyping density is certainly lower in livestock than humans, we think that the very low number of reported non-coding disease-causing alleles is still surprising. We realized that this paragraph was unclearly written in our original manuscript. Our intention was to emphasise the fact that non-coding variants are widely neglected as candidate causal variants, particularly for monogenic traits, and this is due to a lack of functional annotations for non-coding variants. Many studies seem to limit the search space for causal variants to the well-annotated part of the genome. This has now been clarified in the revised manuscript (lines 534-546).

regarding line 353 -consider adding the word potential before fatal since this paper did not demonstrate this.

This has now been rephrased in line 534 onwards in the revised manuscript

Finally, while BioRxiv is a great resource, perhaps the authors should cite the final peer reviewed articles when possible

We have updated citations to peer-reviewed articles wherever possible. Reference 1 has now been changed to the publication in the journal science.

REVIEWERS' COMMENTS:

Reviewer #1 (Remarks to the Author):

The authors of Kadri et al., addressed my previous comments well. We acknowledge the special effort to integrate the splicing QTL studies into the manuscript. These new results provided additional evidences of functional role of predicted branch points. Furthermore, the identification of branch point mutations leading to novel alternative splicing sites confirmed the functional role of branch point, but also it opens the way to the exploration of intronic variants that may be deleterious through alteration of branch point in cattle.

In conclusion, we believe that this new version improved its scientific impact and we recommend this manuscript for publication.

Reviewer #3 (Remarks to the Author):

The revised manuscript has been greatly improved with the inclusion of the addition analyses.

A few minor considerations for the authors to address include: 1) Figure 1A - still does not show the intended colours, 1B - should be colour coded similar to 1C.

2) the inclusion of the testis transcriptomic data does provide a nice demonstration, however it might be prudent for the authors to point out these data may reflect some tissue specificity.

line 304 typo "three"

line 346 include (LD) after linkage disequilibrium sine the acronym is used in the following sentence.

REVIEWERS' COMMENTS:

Reviewer #1 (Remarks to the Author):

The authors of Kadri et al., addressed my previous comments well. We acknowledge the special effort to integrate the splicing QTL studies into the manuscript. These new results provided additional evidences of functional role of predicted branch points. Furthermore, the identification of branch point mutations leading to novel alternative splicing sites confirmed the functional role of branch point, but also it opens the way to the exploration of intronic variants that may be deleterious through alteration of branch point in cattle.

In conclusion, we believe that this new version improved its scientific impact and we recommend this manuscript for publication.

We thank the reviewer for the constructive remarks that helped to greatly improve the manuscript.

Reviewer #3 (Remarks to the Author):

The revised manuscript has been greatly improved with the inclusion of the addition analyses.

A few minor considerations for the authors to address include: 1) Figure 1A - still does not show the intended colours, 1B - should be colour coded similar to 1C.

The intended colour coding was lost in the pdf production, original files for all figures are included with this submission.

We thank the reviewer for the suggested change to Figure 1b, the figure has been modified accordingly.

2) the inclusion of the testis transcriptomic data does provide a nice demonstration, however it might be prudent for the authors to point out these data may reflect some tissue specificity.

We agree with the reviewer. The possible tissue specificity of the alternative splicing is now discussed in line number 365 onwards in the revised manuscript.

line 304 typo "three"

line 346 include (LD) after linkage disequilibrium since the acronym is used in the following sentence.

The suggested changes are made to the revised manuscript